# Visual feedback manipulation in virtual reality to influence pain-free range of motion. Are people with non-specific neck pain who are fearful of movement more susceptible?

**Maaike Kragting**[1,2], **Lennard Voogt**[1,3], **Michel W. Coppieters**[2,4,5]*, **Annelies L. Pool-Goudzwaard**[2,6]

**1** Department of Physical Therapy, Research Centre for Health Care Innovations, Rotterdam University of Applied Sciences, Rotterdam, The Netherlands, **2** Department of Human Movement Sciences, Faculty of Behavioural and Movement Sciences, Amsterdam Movement Sciences, Program Musculoskeletal Health, Vrije Universiteit Amsterdam, Amsterdam, The Netherlands, **3** Pain in Motion Research Group, Department of Physiotherapy, Human Physiology and Anatomy, Faculty of Physical Education and Physiotherapy, Vrije Universiteit Brussel, Brussels, Belgium, **4** Menzies Health Insitute Queensland, Griffith University, Brisbane, Gold Coast, Australia, **5** School of Health Sciences and Social Work, Griffith University, Brisbane, Gold Coast, Australia, **6** Somt University of Physiotherapy, Amersfoort, The Netherlands

* m.coppieters@griffith.edu.au

**Data Availability Statement:** All relevant data are within the manuscript and its Supporting information files.

## Abstract

### Background

Movement-evoked pain may have a protective or learned component, influenced by visual cues which suggest that the person is moving towards a position that may be perceived as threatening. We investigated whether visual feedback manipulation in virtual reality (VR) had a different effect on cervical pain-free range of motion (ROM) in people with fear of movement.

### Method

In this cross-sectional study, seventy-five people with non-specific neck pain (i.e., neck pain without a specific underlying pathology) rotated their head until the onset of pain, while wearing a VR-headset. Visual feedback about the amount of movement was equal, 30% smaller or 30% larger than their actual rotation. ROM was measured using the VR-headset sensors. The effect of VR manipulation in fearful (N = 19 using the Tampa Scale for Kinesiophobia (TSK) and N = 18 using the Fear Avoidance Beliefs Questionnaire-physical activity (FABQ$_{pa}$)) and non-fearful (N = 46; non-fearful on both scales) people was compared using mixed-design ANOVAs.

### Results

Fear of movement, influenced the effect of visual feedback manipulation on cervical pain-free ROM (TSK: p = 0.036, $\eta_p^2$ = 0.060; FABQ$_{pa}$: p = 0.020, $\eta_p^2$ = 0.077); a greater amplitude of pain-free movement was found when visual feedback reduced the perceived rotation angle compared to the control condition (TSK: p = 0.090, $\eta_p^2$ = 0.104; FABQ$_{pa}$: p = 0.030,

**Funding:** The author(s) received no specific funding for this work.

**Competing interests:** The authors have declared that no competing interests exist.

$\eta_p^2 = 0.073$). Independent of the presence of fear, visual feedback manipulation reduced the cervical pain-free ROM in the overstated condition (TSK: $p < 0.001$, $\eta_p^2 = 0.195$; $FABQ_{pa}$: $p < 0.001$, $\eta_p^2 = 0.329$).

## Discussion

Cervical pain-free ROM can be influenced by visual perception of the amount of rotation and people with fear of movement seem to be more susceptible to this effect. Further research in people with moderate/severe fear is needed to determine whether manipulating visual feedback may have clinical applicability to make patients aware that ROM may be influenced more by fear than tissue pathology.

## Introduction

In people with non-specific neck pain (i.e., neck pain without a specific underlying pathology [1]), movement-evoked pain may be a protective or learned response. This response may be influenced by visual cues which suggest that a patient is moving their head towards a position that may be perceived as threatening [2]. Visual cues previously accompanied by pain during certain cervical movements may be stored in memory and may become associated with the perception of pain [3–5]. Results from a previous study [2] support this hypothesis, by showing that cervical pain-free range of motion can be influenced in people with non-specific neck pain by altering visual feedback regarding the amount of rotation in a virtual reality (VR) environment. This finding is interesting, as the use of VR may facilitate people with neck pain to learn a new association (i.e., a previously painful movement becomes pain-free), which can be of therapeutic value. However, a large variability between participants existed regarding the size of the effect of visual feedback manipulation on changes in pain-free range of motion [6]. Another study using a similar methodology [2] did not find an effect of visual feedback manipulation on cervical pain-free range of motion in a larger group of people with non-specific neck pain [7]). Hence, further research into the role of visual stimuli on pain perception is needed.

A possible explanation for the different results between studies regarding the role of visual stimuli on pain perception might be that pain is a conditioned response in only a subgroup of patients. Following the assumption that visual cues serve as warning signals associated with pain that prevent people from moving beyond a certain, learned, movement range [2, 8, 9], it is hypothesized that individuals who report high levels of fear of movement may be more prone to visual feedback manipulation than others. The underlying assumption is that individuals who are not fearful of movement may unlearn the association made between movement and pain, allowing pain to be reduced. In contrast, in individuals who are fearful of movement these maladaptive patterns of learning may continue to exist because of avoidance behaviour [8, 10–14].

The aim of the current study was to evaluate whether the effect of visual feedback manipulation was different in people with non-specific neck pain who are fearful of movement, compared to people with non-specific neck pain who are not fearful of movement.

## Methods and materials

### Participants

People with non-specific neck pain were recruited between April and December 2018 from 12 primary care physiotherapy clinics in The Netherlands, using the convenience sample method.

The physiotherapist performed the initial screening. Inclusion criteria were: 1) neck pain Grade I and II (i.e., Grade I: neck pain with no signs of major pathology and no or little interference with daily activities; Grade II: neck pain with no signs of major pathology, but with interference with daily activities [15]) that was provoked and/or aggravated by cervical rotation, 2) aged between 18–65 and 3) being able to read and understand Dutch. People were excluded if they had neck pain with neurological signs (i.e., neck pain Grade III), neck pain with signs of serious pathology (i.e., neck pain Grade IV) [15] or if their vision was impaired (e.g., people who had poor vision when not wearing their glasses as they could not be worn in the VR condition. The use of contact lenses was not a problem).

A sample size calculation was carried out *a priori* using an ANOVA repeated-measures within-between interaction in G-power 3.1 [16]. Because of the mixed design that was used in this study, we expected a smaller effect than the (within) effect of altered visual feedback on cervical pain-free range of motion as revealed in a previous study ($\eta_p^2 = 0.29$) [2]. Therefore, we devided this effect by two, which was an arbitrary choice. Based on an expected effect size of $\eta_p^2 = 0.145$ (i.e., 0.29/2), a significance level of $\alpha < 0.05$, a power of 0.8, 2 groups, 3 measurements and assuming a 75% correlation among repeated-measures, the minimum required number of participants was 40, i.e., 20 per group.

The Scientific and Ethical Review Board (VCWE) of the Vrije Universiteit Amsterdam approved the study (VCWE-2016-218R1). Prior to study participation, all participants received an information letter and provided written consent. The STROBE reporting guideline for cross-sectional observational studies was followed [17].

## Procedure

Participants completed a digital questionnaire (Qualtrics, Provo, UT) to collect personal and neck pain related information regarding age, sex, the duration and onset of their neck pain (gradual or sudden, and if sudden, history of trauma), kinesiophobia, fear of physical activity, neck pain intensity and disability (see 'Measurements' section below for further specifications). Within a week after completion of the questionnaire, the VR-experiment was performed at one of the participating physiotherapy practices, to evaluate the effect of visual feedback manipulation on cervical pain-free range of motion. Motion sickness is a common side effect in VR [18], especially in people with neck pain [19], and a possible barrier when considering implementing VR in clinical practice. Therefore, motion sickness was evaluated following the VR-experiment using the short version of the Misery Scale (sMISC).

## Measurements

**Fear.** Fear is a complex multidimensional phenomenon containing several distinct, but closely related constructs, such as fear of movement/reinjury, kinesiophobia, fear of pain and fear avoidance [20]. Congruent with our hypothesis and taking into account the multidimensionality of the construct fear, we used two different questionnaires to create the fear of movement subgroups (i.e., people with either kinesiophobia or fear of physical activity, or both), being the (1) Tampa Scale of Kinesiophobia (TSK) (construct: kinesiophobia) and (2) Fear Avoidance Beliefs Questionnaire, physical activity subscale ($FABQ_{pa}$)) (construct: fear of physical activity). The TSK and the $FABQ_{pa}$ are validated self-reported instruments recommended to assess fear of movement [20, 21]. Both scales use standardised cut-off points to distinghuish people with and without fear of movement.

The TSK consists of 17 items, each scored on a 4-point Likert scale [22, 23]. The total score ranges from 17 to 68, with a score >37 being regarded as indicative for kinesiophobia [23]. The Dutch version of the TSK is reliable and has been validated in people with low back pain

and fibromyalgia [24], but has not yet been validated for neck pain. The psychometric properties of the TSK in people with neck pain have been studied in several countries in various neck pain populations. Reliability and validity are moderate to good [25–27].

The FABQ$_{pa}$ contains 4 items regarding beliefs about how physical activity affects pain, each scored on a 6-point Likert scale [28]. A score >14 is indicative for fear of physical activity [29, 30]. The internal consistency, test-retest reliability and validity of the FABQ in people with neck pain is acceptable [25, 26, 31].

**Pain and disability.**   Pain intensity was measured on a 11-point scale using the Numeric Pain Rating Scale (NPRS), which is simple and valid tool [32]. The Neck Disability Index (NDI) was used to assess the level of disability in people with neck pain. It is a reliable and responsive tool which consists of 10 items with six response categories (range 0–5, total score range 0–50), with higher scores representing higher disability [33–35].

**Motion sickness.**   The Misery Scale (short version) (sMISC) was used to assess feelings of misery [36–38]. This scale is easy to administer and correlates strongly with the more extensively validated Simulator Sickness Questionnaire [36, 39]. The sMISC is scored on a 6-point scale in which 0 = No symptoms, 1 = Mild symptoms, but no nausea, 2 = Severe symptom, but no nausea, 3 = Mild nausea, 4 = Severe nausea and 5 = Vomiting.

### Visual feedback manipulation and cervical pain-free range of motion

During the VR-experiment participants sat on a chair, wearing a fixation belt to limit upper trunk movement (see Fig 1). They wore a VR-headset (Oculus Rift head-mounted display; Oculus VR, Irvine, CA, United States), and noise cancelling headphones to reduce ambient noise and to distract them from the research setting by playing background music (nature sounds), which was held constant between the conditions. Standardised instructions on what tasks to perform were delivered via the headphones using pre-recorded audio files.

Participants faced forward and were asked to rotate their head slowly to the left until the onset of pain, then to the right until the onset of pain, and then back to the midline, while submerged in a VR environment. This rotation was repeated six times in three different conditions. In two conditions the illusion was created of moving 30% less (understated visual feedback) or moving 30% more (overstated visual feedback) than the actual physical rotation. A condition with accurate visual feedback was considered the control condition. A total of eighteen rotations were performed. To create an illusion of a smaller or larger movement than the actual movement, a technique called 'redirected walking' was used [40]. This technique tracks the actual head rotation and transforms it to the VR environment in an overstated or understated form. The rotation gain (i.e., the factor by which actual rotation translates to the visual rotation as shown by the VR headset ($Gain_{rot} = Rot_{virtual}/Rot_{real}$) (2), was set at 0.7 (understated condition), 1.0 (control condition) and 1.3 (overstated condition). The choice for the 0.7 and 1.3 gains was based on the results of a pilot study that aimed to determine the largest rotation gains that were more likely to be judged as 'not manipulated' than 'manipulated'. For more detailed information on this pilot study, please see S1 Appendix.

There was a two-minute rest period after six repetitions to minimise motion sickness and to assess pain intensity during the experiment, as fluctuations in pain may influence the results. During the rest period participants kept the VR-headset on and had their eyes closed.

The total range of motion (i.e., from maximal left rotation to maximal right rotation) for each repetition was measured in degrees using the sensors of the VR-headset which was connected to a computer running Windows 10 (Microsoft Corporation, Redmond, WA, USA). This method of data acquisition and its validation has been described in detail in a previous publication [7]. Subsequently, the average range of motion of six repetitions was calculated for

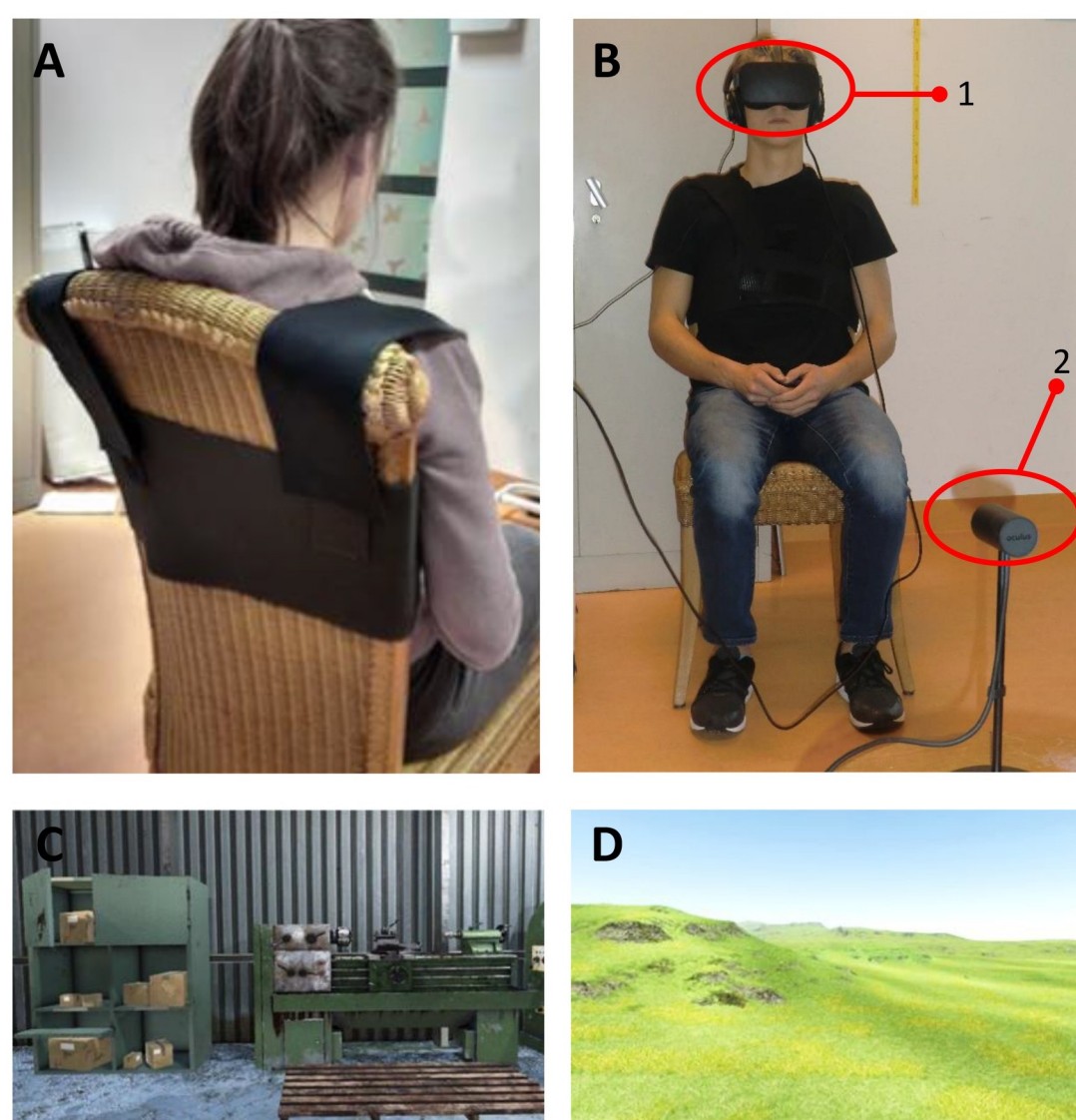

**Fig 1. Measurement setup during the VR-experiment. A**. The participant sat on a chair and wore a fixation belt over the shoulders and upper torso to prevent trunk rotation. **B**. The participant wore the VR-headset (1) and sat in front of the Oculus-sensor (2). **C** and **D**. Examples of the Virtual Reality Environments projected in the VR-headset. Please note that the current study used the same materials (e.g., VR-headset, chair, fixation belt) in an equivalent setting as a previous study [7]. However, both the aims of the studies and the participants differed.

each gain condition (i.e., absolute data). To account for differences in the overall neck range of motion between subjects, for each participant, the data from the overstated and understated condition were transformed to a proportion of the mean range of rotation in the control condition (i.e., relative data). Between subject differences (especially between people with and without fear of movement) were anticipated [41, 42].

The order in which the gain conditions were offered (increasing gains (i.e., 0.7; 1.0; 1.3) or decreasing gains (i.e., 1.3; 1.0; 0.7)) was randomised and counterbalanced across participants. We used block randomization to balance the distribution between increasing and decreasing gains throughout the study. Participants were blinded for the sequence of each condition. The experimenter was blinded for the presence of fear. For each gain condition, six different

environments (loft, workman's shed, living room, dungeon, undulating landscape and a forest) (Unity 5.3.1, Unity Technologies, San Francisco, CA, United States) were used to prevent the participants from recalling their previous position. Hence, one environment was used per repetition for each condition. We chose to use the same types of (indoor and outdoor) environments as in a previous study [2] (see S1 Fig).

At the end of the experiment, participants were asked whether they had noticed any differences between the multiple rotations to assess whether they were naïve to the gain change.

## Statistical analyses

To examine whether the effect of visual feedback manipulation on cervical pain-free range of motion was different between people with fear and people without fear, two separate analyses were performed (one for TSK, and one for FABQ$_{pa}$), using the relative data (reported in the main article) and the absolute data (reported in Table 2 and the S2 Appendix). For both the relative and absolute data, people were classified separately as being fearful according to the TSK and FABQ$_{pa}$ scores (TSK: fearful for movement (kinesiophobia), i.e., TSK>37; FABQ$_{pa}$: fear of physical activity, i.e., FABQ$_{pa}$>14), or non-fearful for movement (i.e., negative on both scales; i.e., TSK≤37 & FABQ$_{pa}$≤14). In both analyses, a General Linear Model (GLM), repeated-measures mixed-design ANOVA (with the within-variable gain (3) and between-variable fear (2)) was performed [43]. The results were presented using exact p-values, the effect estimate and its 95% confidence interval. Partial $\eta^2$ was calculated to determine the effect size. An effect size between 0.01 and 0.059 was considered small, between 0.06 and 0.138 medium and ≥0.139 was considered large [44].

The assumption of normality was assessed by visual inspection of the histograms and Q-Q plots. The assumption of equality of variances between the two subgroups (fear/no fear) per analysis on cervical pain-free range of motion in the understated and overstated gain conditions was checked using Levene's test for homogeneity of variance. The assumption of sphericity was checked according to Girden [45]. When appropriate, the mixed-design ANOVA was followed-up with simple contrasts to identify whether specific differences occurred between the three gain conditions.

Pain ratings were compared among conditions using a repeated-measures ANOVA or, in case of violations of the normality assumptions, a Friedman's ANOVA [46, 47]. For the ordinal sMISC scores, frequencies and percentages were reported. All statistical analyses were conducted using SPSS (IBM Corp. (2020). IBM SPSS Statistics for Windows, Version 27.0. Armonk, NY).

## Results

### Participant characteristics and comparability between groups

Seventy-five people with neck pain volunteered for the study (51 females; mean (SD) age: 44.3 (14.5) years)). One additional participant was unable to complete the experiment due to nausea and was excluded from the study. We decided to recruit more than the 40 participants that were required based on the sample size calculation, because the subgroups (fear/no fear) were not equally distributed. We enrolled participants until we had sufficient people with fear of movement.

Using the TSK score, 19 participants had kinesiophobia. Using the FABQ$_{pa}$ score, 18 participants had fear of physical activity. Forty-six participants had no form of fear of movement (TSK≤37 & FABQ$_{pa}$≤14). The TSK score was missing for one participant and the FABQ$_{pa}$ score was missing for two participants. These participants were excluded from this analysis. In both analyses, there were no differences between the fearful and non-fearful group in age and

duration of the neck pain, but the fearful group felt more disabled, scored higher on pain intensity and was more limited in the cervical pain-free range of rotation. Table 1 provides an overview of the participant characteristics per group including the results from the comparisons between subgroups.

**Table 1. Participant characteristics and comparisons between groups.**

| Variables | Total N = 75 | Subgroups | | | Mean differences between the No fear of movement and Kinesiophobia subgroups (t-test/MannW) | Mean differences between the No fear of movement and Fear of physical activity subgroups (t-test/MannW) |
|---|---|---|---|---|---|---|
| | | No fear of movement (TSK$\leq$37 & FABQ$_{pa}$ $\leq$14) N = 46[ab] | Kinesio phobia (TSK>37) N = 19[a] | Fear of physical activity (FABQ$_{pa}$ >14) N = 18[b] | | |
| Women (N (%)) | 51 (68%) | 33 (72%) | 11 (58%) | 11 (61%) | | |
| Trauma (car-accident or fall) in history | 23 (31%) | 12 (26%) | 9 (47%) | 6 (33%) | | |
| Stage of disorder | N = 71[c] | N = 44 | N = 19 | N = 17 | | |
| Acute (0–3 weeks) | 4 (6%) | 2 (5%) | 1 (5%) | 2 (12%) | | |
| Subacute (3 weeks-3 months) | 16 (22%) | 10 (23%) | 4 (21%) | 4 (24%) | | |
| Chronic (>3–24 months) | 27 (38%) | 16 (36%) | 8 (42%) | 5 (29%) | | |
| Long-lasting chronic (>24 months) | 24 (34%) | 16 (36%) | 6 (32%) | 6 (35%) | | |
| Age (in years) (Mean (SD)) | 44.3(14.5) | 45.4 (13.8) | 46.8 (14.3) | 45.7 (16.0) | $t_{(63)} = -0.381$, p = 0.704 | $t_{(62)} = -0.068$, p = 0.946 |
| Duration of neck pain (in months)[c] | 12 (2.5–72) | 13.5 (2.6–102) | 12 (3–60) | 8 (2.3–42) | †U = 378.00, z = -0.60, p = 0.549 | †U = 320.00, z = -0.87, p = 0.385 |
| Pain intensity (0–10 NPRS) | | | | | | |
| Average last week (Mean (SD)) | 5.0 (2.1) | 4.2 (2.0) | 6.6 (1.4) | 6.2 (1.8) | $t_{(63)} = -4.644$ p< 0.001 | $t_{(62)} = -3.692$ p< 0.001 |
| Maximum last week (Mdn (25–75%)) | 7.0 (5.0–8.0) | 6.0 (3.8–8.0) | 8.0 (7.0–9.0) | 8.0 (7.0–9.0) | †U = 129.00, z = -4.45 p< 0.001 | †U = 167.50, z = -3.72 p< 0.001 |
| Disability (0–50 NDI) (Mdn 25–75%)) | 13 (7.0–19.0) | 9.5 (6.0–14.0) | 19 (7.0–26.0) | 17.5 (12.8–24.5) | †U = 129.00, z = -4.45, p< 0.001 | †U = 185.00, z = -3.43, p = 0.001 |
| Pain-free Cervical Range of Motion[d] (degrees (SD)) | 120.9 (31.4) | 128.5 (24.9) | 96.5 (32.4) | 106.9 (38.4) | $t_{(63)} = 4,302$ p< 0.001 | $t_{(22,8)} = 2.211$ p = 0.037 |
| Kinesiophobia (TSK) (Mean (SD)) | 33.5 (7.0) | 29.7 (4.8) | 42.2 (4.2) | 39.6 (5.7) | $t_{(63)} = -12.493$, p< 0.001 | $t_{(62)} = -7.077$, p< 0.001 |
| Fear of physical activity (FABQ$_{pa}$) (Mean (SD)) | 10.2 (5.0) | 7.6 (4.0) | 14.3 (3.8) | 16.4 (1.7) | $t_{(63)} = -6.211$, p< 0.001 | †U = 0.00, z = -6.20, p< 0.001 |

[a] For one participant, the Tampa score was missing. Therefore, this participant was not included in one of the subgroups.

[b] For two participants, the FABQ$_{pa}$ score was missing. Therefore, these participants were not included in one of the subgroups.

[c] For four participants, the duration of the neck pain was missing.

[d] For the pain-free Cervical Range of Motion the mean scores in the control condition are used.

N: number; SD: standard deviation; Mdn: Median; NDI: Neck Disability Index; NPRS: Numeric Pain Rating Scale; TSK: Tampa Scale for Kinesiophobia; FABQ$_{pa}$: Fear Avoidance Beliefs Questionnaire—physical activity; t-test/MannW: Mean difference according to a (parametric) independent Samples T-Test, or a †(non-parametric) Mann-Whitney U test.

**Table 2. Influence of visual feedback manipulation on cervical pain-free range of motion.**

| Gain condition | Total range of motion (degrees) (mean [95%CI]) | | | |
|---|---|---|---|---|
| | All participants (N = 75) | No fear of movement (TSK≤37 & FABQ$_{pa}$≤14) (N = 46[*/**]) | Kinesiophobia (TSK>37) N = 19[*] | Fear of physical activity (FABQ$_{pa}$>14) N = 18[**] |
| **Relative data[1]** | | | | |
| 0.7 gain | 1.008 [0.991, 1.025] | 0.993 [0.978, 1.009] | 1.043 [0.996, 1.090] | 1.036 [0.985, 1.088] |
| 1.0 gain | 1.0 [1.0, 1.0] | 1.0 [1.0, 1.0] | 1.0 [1.0, 1.0] | 1.0 [1.0, 1.0] |
| 1.3 gain | 0.964 [0.948, 0.981] | 0.969 [0.951, 0.987] | 0.955 [0.910, 1.000] | 0.935 [0.901, 0.969] |
| **Absolute data[2]** | | | | |
| 0.7 gain | 120.8 [114.0, 127.5] | 127.3 [120.1, 134.4] | 99.5 [84.4, 114.6] | 108.7 [91.0, 126.4] |
| 1.0 gain | 120.9 [113.7, 128.1] | 128.5 [121.1, 135.9] | 96.5 [80.8, 112.1] | 106.9 [87.8, 126.0] |
| 1.3 gain | 117.0 [109.5, 124.4] | 124.7 [116.9, 132.5] | 92.3 [76.6, 107.9] | 101.2 [81.6, 120.8] |

[1] Relative data: a proportion of the mean cervical range of rotation in the control condition.

[2] Absolute data: the total cervical range of motion (i.e., the sum of left and right rotation) in degrees

* For one participant, data regarding the Tampa score was missing. Therefore, this participant was not included in one of the subgroups.

** For two participants, the FABQ$_{pa}$ score was missing. Therefore, these participants were not included in one of the subgroups.

N: number; 95%CI: 95% Confidence Intervals [lower bound, upper bound]

The differences in absolute range of motion between the groups were large. The mean (SD) cervical pain-free range of rotation in the control condition in people with kinesiophobia (TSK) was 96.5 (32.4) degrees, in people with fear of physical activity 106.9 (38.4) degrees and in the 'no fear' group 128.5 (24.9) degrees (See Tables 1 and 2 for further specification of absolute range of motion differences between the groups). As this could influence the change induced by visual manipulation as a function of their total cervical ROM, in further analyses, we took into account the relative change in range of motion.

## Statistical assumptions

Based on visual inspection of the Q-Q plots and the histograms, the relative data were considered sufficiently normally distributed. For both analyses (using the TSK score and the FABQ$_{pa}$ score), Levene's test revealed that there were no violations of the assumption of homogeneity of variance between the gain conditions. Sphericity was violated for both analyses and the Greenhouse-Geisser EPSILON was <0.75, therefore the Greenhouse-Geisser correction was used in both analyses [46].

## Effect of fear and visual feedback manipulation on cervical pain-free range of motion

Using the relative data, the ANOVA revealed an interaction effect between fear of movement (as determined by the TSK and the FABQ$_{pa}$ subscale) and visual feedback manipulation on cervical pain-free range of motion with a medium effect size (TSK: p = 0.036, $\eta_p^2$ = 0.060; FABQ$_{pa}$; p = 0.020, $\eta_p^2$ = 0.077). This indicates that cervical pain-free range of motion when the visual feedback was manipulated depended on the presence of fear of movement (see Fig 2A & 2B and Table 2). Contrasts revealed that the cervical pain-free range of motion in people with fear was larger in the understated condition, with a medium effect size, compared to the control condition, while not in people without fear of movement (TSK: p = 0.009, $\eta_p^2$ = 0.104;

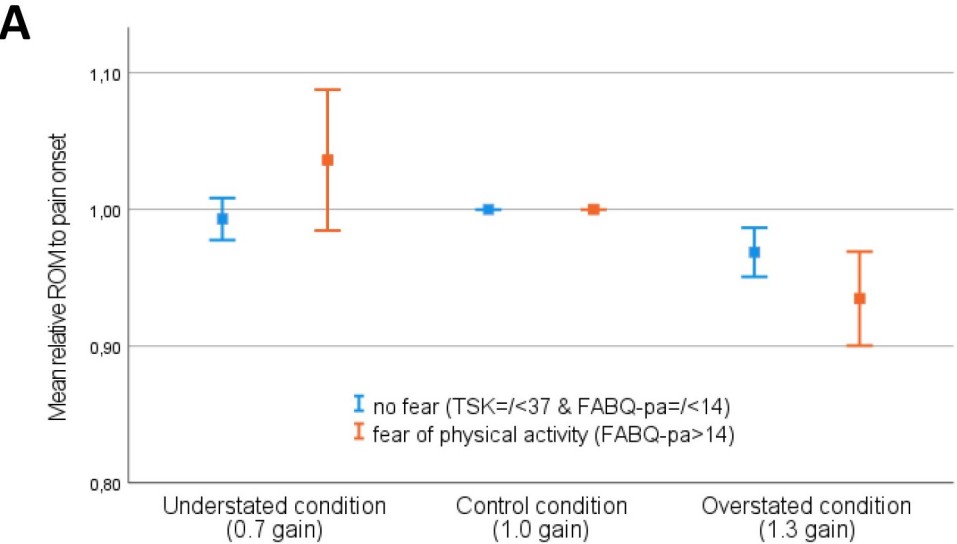

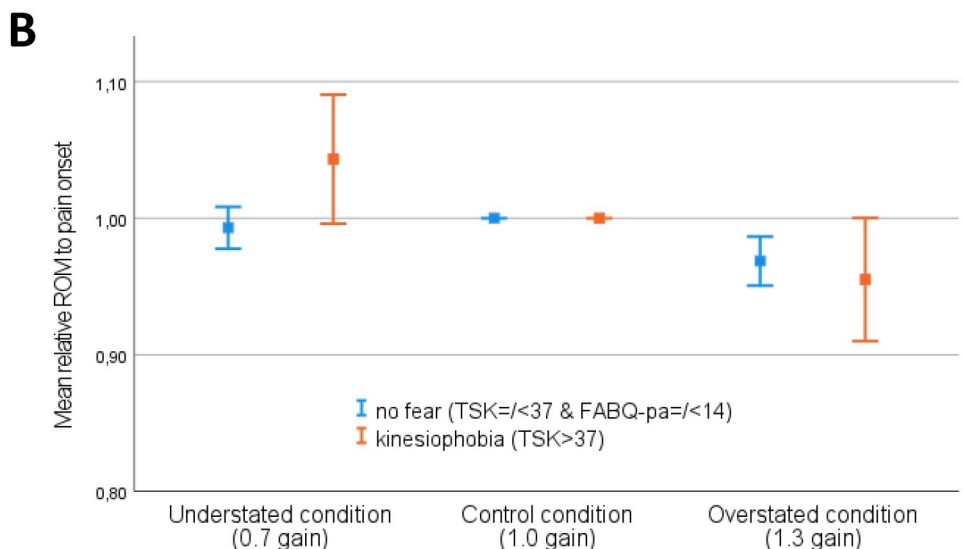

**Fig 2. Effect of visual feedback manipulation on cervical pain-free range of motion.** Effect of visual feedback manipulation on cervical pain-free range of motion in people without fear of movement (Tampa$\leq$37 & FABQ$_{pa}\leq$14) versus people with kinesiophobia (Tampa>37) (A) and versus people with fear of physical activity (FABQ$_{pa}$>14) (B). Please note that the relative change in range of motion is used (i.e. a proportion of the mean range of rotation in the control condition). The error bars represent the 95% confidence intervals.

FABQ$_{pa}$: p = 0.030, $\eta_p^2$ = 0.073). In the overstated condition compared to the control condition, the difference in the cervical pain-free range of motion between people with and without fear was small (TSK: p = 0.491, $\eta_p^2$ = 0.008; FABQ$_{pa}$; p = 0.057, $\eta_p^2$ = 0.057). More specifically, in people with fear of movement pain-free range of motion increased by 4.3% (95%CI: -0.4%, 9.0%) (when classified on TSK scores) and by 3.6% (95%CI:-1.5, 8.8%) (when classified on FABQ$_{pa}$ scores) in the understated condition compared to the control condition, and decreased by -4.5% (95%CI: -9.0%, 0.0%) (TSK) and -6.5% (95%CI:-9.9%, -3.1%) (FABQ$_{pa}$) in

the overstated condition compared to the control condition. In people without fear, the effect of visual feedback manipulation was only present in the overstated condition: pain-free range of motion decreased by -3.1% (95%CI: -4.9%, 1.3%) in the overstated condition compared to the control condition. The overall effect of the manipulation (i.e., the difference in cervical pain-free range of motion between the understated and the overstated condition) in the kinesiophobia group was 8.8% (95%CI: 8.6%, 9.0%), and 10.1% (95%CI: 8.4%, 11.9%) in the fear of physical activity subgroup, versus 2.4% (95%CI: 2.2%, 2.7%] in the no-fear group.

Further analyses showed a main effect of gain (i.e., independent of the presence of fear) on cervical pain-free range of motion with a large effect size (TSK: $p<0.001$, $\eta_p^2 = 0.158$; FABQ$_{pa}$: $p<0.001$, $\eta_p^2 = 0.195$). Within subjects contrasts showed that this effect of gain was large for the overstated condition compared to the control condition (TSK: $p< 0.001$, $\eta_p^2 = 0.195$; FABQ$_{pa}$: $p<0.001$, $\eta_p^2 = 0.329$), while small for the understated condition versus the control condition (TSK: $p = 0.055$, $\eta_p^2 = 0.057$; FABQ$_{pa}$: $p = 0.137$, $\eta_p^2 = 0.035$). For the results of the analyses using the absolute data, see Table 2, S3 Fig in S2 Appendix.

### Exploratory analysis

The results showed that the mean TSK and FABQ$_{pa}$ scores in the 'fear groups' were close to the cut-off points to be considered fearful. To increase the contrast between the 'no fear' and 'fear' group, we decided to perform an exploratory analysis in which people who were negative on both scales (N = 46; TSK$\leq$37 & FABQ$_{pa}\leq$14) were compared to participants who scored positive on both scales (N = 10; TSK>37 & FABQ$_{pa}$>14). The effect size regarding the interaction effect between fear of movement and visual feedback manipulation on cervical pain-free range of motion was then twice as large ($p = 0.007$, $\eta_p^2 = 0.112$). In people with fear of movement cervical range of motion increased by 5.2% (95%CI: -2.1%, 12.4%) in the understated condition and decreased by -7.5% (95%CI:-12.6%, -2.4%) in the overstated condition. The overall effect of the manipulation (i.e., the difference in cervical pain-free range of motion between the understated and the overstated condition) in the fear group was 12.7% (95%CI: 10.5%, 14.8%). The results of the exploratory analysis are included in S1 Table.

### Pain intensity

Due to the non-normal distribution in the pain scores, a Friedman's ANOVA was performed to compare pain intensity between the three gain conditions (Median (IQR 25–75%): 4 (2.0, 6.0) in the 0.7 gain condition, 4 (3.0–7.0) in the 1.0 gain condition and 4 (2.75–6.0) in the 1.3 gain condition). There was no significant difference in pain intensity in different gain conditions ($\chi^2(2) = 1.39$, $p = 0.499$).

### Awareness of gain change

Sixty-two percent of all participants (46/74; the score of one participant was missing) reported not to have noticed a change in gain between the conditions (63% (N = 12/19) in the FABQ$_{pa}$ fear group; 61% (n = 11/18) in the TSK fear group).

### Motion sickness

During the VR submersion, the participants experienced either no symptoms (65%), mild symptoms but no nausea (17%), severe symptoms but no nausea (4%), mild nausea (8%) or severe nausea (5%) (see Table 3).

**Table 3. Misery scores at the end of the experiment.**

| sMISC scores | All participants (N = 75) | No fear of movement (TSK≤37 & FABQ$_{pa}$≤14) (N = 46)*/** | Kinesiophobia (TSK>37) (N = 19)* | Fear of physical activity (FABQ$_{pa}$>14) (N = 18)** |
|---|---|---|---|---|
| 0: No Nausea or other symptoms | 49 (65.3%) | 29 (63.0%) | 11 (57.9%) | 15 (83.3%) |
| 1: Mild symptoms, but no nausea | 13 (17.3%) | 8 (17.4%) | 4 (21.1%) | 1 (5.6%) |
| 2: Severe symptoms, but no nausea | 3 (4.0%) | 1 (2.2%) | 2 (10.5%) | 1 (5.6%) |
| 3: Mild nausea | 6 (8.0%) | 6 (13.0%) | 0 (0.0%) | 0 (0.0%) |
| 4: Severe nausea | 4 (5.3%) | 2 (4.3%) | 2 10.5%) | 1 (5.6%) |
| 5: Vomiting | 0 (0.0%) | 0 (0.0%) | 0 (0.0%) | 0 (0.0%) |

Data are reported as number of patients (%).

* For one participant, data regarding the Tampa score was missing. Therefore, this participant was not included in one of the subgroups.

**For two participants, the FABQ$_{pa}$ score was missing. Therefore, these participants were not included in one of the subgroups.

## Discussion

This study revealed that fear of movement (as assessed by the TSK and the FABQ$_{pa}$ subscale) influenced the susceptibility for the effect of visual feedback manipulation in people with non-specific neck pain. Regardless of whether people were fearful or not, this study confirmed that cervical pain-free range of motion can be influenced by visual perception of the amount of rotation. However, in people with fear of movement, the cervical pain-free range of motion was larger in the understated feedback condition compared to the control condition, and was smaller in the overstated feedback condition, while in people without fear, there was only a decrease in cervical pain-free range of motion in the overstated feedback condition. Furthermore, the difference in cervical pain-free range of motion between the understated and the overstated condition was larger in people with fear of movement compared to people without fear (i.e., a medium effect).

The age and sex of the participants in this study is representative for people with non-specific neck pain [48]. People with fear of movement were a distinguishable subgroup of the participants with neck pain, with higher scores for pain and dysfunction and a more limited cervical pain-free range of motion. This finding is consistent with previous findings [41, 42, 49–51], although correlations between fear and range of motion are inconsistent [41, 42, 52, 53]. Fearful people might share some similarities with people with anxiety disorders. Several studies in people with anxiety disorders showed deficits in classical conditioning, such as over-generalisation (i.e., the process in which safe stimuli, which share some characteristics of the threatening conditioned stimulus, also lead to threat) [14, 54, 55]. The ability to discriminate between safe and threatening stimuli is conditional to avoid generalization. Differences between people with and without neck pain were found in the ability to distinguish 'threatening' cues from 'non-threatening' cues [56]. This suggests that maladaptive associative learning might contribute to disability in people with neck pain who are fearful for movement.

In the current experiment we assumed that, based on associative learning, nociceptive stimuli and (non-nociceptive) visual stimuli had been coupled, and that the conditioned response (i.e., movement-evoked pain) could be influenced by manipulating visual feedback [2)] especially in fearful people. We found an effect of visual feedback manipulation on cervical pain-free range of motion and this effect was larger in people with fear of movement. In people who were fearful, the direction of the effect of visual feedback manipulation (i.e., a greater amplitude of pain-free movement in the understated condition versus a smaller amplitude in

the overstated condition) was as expected. In people without fear, we did not find a greater amplitude of cervical pain-free range of motion when visual feedback reduced the perceived rotation angle compared to the control condition. This finding supports the idea that movement evoked pain has a protective or learned component in people with fear of movement. Our results in the fear groups regarding the effect of visual feedback manipulation were consistent with a previous study [2]. They also found an increase in range of motion in the understated condition (6%, 95%CI: 2%, 11%) and a decrease in range of motion in the overstated condition (7%, 95%CI: 3%, 11%) in people with neck pain (p<0.001, $\eta_p^2$ = 0.39). However, participant selection in that study (N = 24) was not based on the presence of fear, the gain change was 20% and the influence of fear was not studied. Another study in a larger sample (N = 75) of people with neck pain, that used a similar design and gain conditions of 0.8 and 1.2, found a small effect of visual feedback manipulation on cervical pain-free range of motion (p = 0.133, $\eta_p^2$ = 0.031) [7]. The role of fear was also not investigated in that study. This suggests that the susceptibility to modified visual feedback, may depend on the presence of fear of movement, and on the VR characteristics used. Based on the results in the current study and a comparison with previous studies [2, 7, 57] we consider that the magnitude of the gain change and alternating different VR environments may be important. The type of VR-environment seems less important, as another study shows that pain sensitivity was not modulated by VR-environment [58].

The lack of a strong interaction effect between kinesiophobia or fear of physical activity and visual feedback manipulation on cervical pain-free range of motion might have been due to the fact that (1) the sample of the group of fearful people was relatively small and (2) the contrast between people who were 'fearful' and 'not fearful' was relatively small. Mean TSK and FABQ$_{pa}$ scores in the 'fear group' were close to the cut-off points to be considered fearful. People in the 'fear group' experienced mild/moderate fear, which is typical for people with neck pain in a primary care setting [59]. The results of our exploratory analysis, in which we increased the contrast between the groups by comparing participants who scored positive on both scales (N = 10: TSK≤37 & FABQ$_{pa}$≤14) with people with 'no fear' (N = 46), confirmed the impact of fear as the effect size was twice as large. However, due to the small number of people who scored positive on both questionnaires, these results should be interpreted with caution.

Although there was an effect of visual feedback manipulation on cervical pain-free range of motion independent of fear, in people who were not fearful of movement this effect was limited to the overstated condition and the change in range of motion between the understated and the overstated visual feedback condition was small (participants reduced their range of motion with -2.4% (95%CI: -2.2%, -2.7%)) in the overstated condition compared to the understated condition). We believe that for this group using overstated visual feedback to influence pain perception, may have little clinical utility. First, the absolute difference in total range of motion between the understated and overstated condition found in this study (2.6 degrees in the 'no fear' group) will not be detectable in a clinical setting without the use of accurate motion sensors. Second, no increase in cervical pain-free range of motion was found in the understated condition, while this is typically pursued in a therapeutic setting. Furthermore, a pilot study in 8 people with chronic neck pain, that was based on the hypothesis that overstated visual feedback could be used to reduce the association made between movement and pain (following the extinction principle) did not show that this reduced pain, even while VR was used for a longer period (21–28 days) [60]. In contrast, it might be that the use of VR has clinical relevance for people with moderate/severe fear of movement. Visualizing the relatively large change in cervical pain-free range of motion between the understated and overstated visual feedback condition (12.7%, 95%CI: 10.5%, 14.8%) in people with fear on both the TSK

& FABQ$_{pa}$) may contribute to people's understanding that non-nociceptive stimuli play a role in their pain experience. Moreover, exercising in an understated feedback condition might facilitate the use of a larger range of motion and influence the association made between movement and pain. Further research in which VR is applied with a large gain (i.e., a minimum of 30% gain) seems interesting to investigate whether this actually produces clinically relevant results in people with moderate/severe fear.

This experiment supports the view that cervical pain-free range of motion in rotation can be influenced by visual perception of the amount of rotation, in people with non-specific neck pain. People with fear of movement seem to be more susceptible for the effect of visual feedback manipulation than people without fear of movement. Because of the limitations mentioned, the results should be interpreted with caution.

## Supporting information

**S1 Appendix. Pilot study 'creating illusions: Perception of gain changes when manipulating the visual feedback with virtual reality'.**
(DOCX)

**S2 Appendix. Results of the data-analyses using the absolute data (range of rotation in degrees).**
(DOCX)

**S1 Fig. Examples of the virtual reality environments projected in the VR-headset.**
(DOCX)

**S1 Table. Exploratory analysis: Influence of visual feedback manipulation on cervical pain-free range of motion in people who scored positive or negative on both scales.**
(DOCX)

**S1 Dataset.**
(SAV)

## Acknowledgments

The authors would like to thank the following students for their assistance in data collection: Theodora Karagianni, Sanne Akkermans, Sophie Boelen, Manouk Boer, Aytac Cakir, Thouria Loukili, Mandy Andriessen, Lisa Verhagen, Jorjen Neumann, Suzanne ten Hoor, Freek van der Velden, Jan van Splunter, Joey van Eeden, Laurens Sluiter and Ralph Laenen. This research did not receive financial support from funding agencies in the public, commercial, or not-for-profit sectors.

## Author Contributions

**Conceptualization:** Maaike Kragting, Lennard Voogt, Michel W. Coppieters, Annelies L. Pool-Goudzwaard.

**Data curation:** Maaike Kragting.

**Formal analysis:** Maaike Kragting, Michel W. Coppieters.

**Investigation:** Maaike Kragting.

**Methodology:** Maaike Kragting, Lennard Voogt, Michel W. Coppieters, Annelies L. Pool-Goudzwaard.

**Project administration:** Maaike Kragting.

**Resources:** Maaike Kragting.

**Supervision:** Lennard Voogt, Michel W. Coppieters, Annelies L. Pool-Goudzwaard.

**Validation:** Annelies L. Pool-Goudzwaard.

**Visualization:** Maaike Kragting, Michel W. Coppieters.

**Writing – original draft:** Maaike Kragting.

**Writing – review & editing:** Maaike Kragting, Lennard Voogt, Michel W. Coppieters, Annelies L. Pool-Goudzwaard.

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
