## [Decision Letter · Decision Letter 0]

14 Mar 2023

PONE-D-22-28987Visual feedback manipulation in virtual reality to influence pain-free range of motion. Are people with pain who are fearful of movement more susceptible?PLOS ONE

Dear Dr. Kragting,

Thank you for submitting your manuscript to PLOS ONE. After careful consideration, we feel that it has merit but does not fully meet PLOS ONE’s publication criteria as it currently stands. Therefore, we invite you to submit a revised version of the manuscript that addresses the points raised during the review process.

We look forward to receiving your revised manuscript.

Kind regards,

Mariella Pazzaglia

Academic Editor

PLOS ONE

Additional Editor Comments:

I have invited comments from experts from your research domain.

As you will see, the paper is interesting, but design and method would need to be addressed/explained carefully.

Taken altogether, let me invite you to prepare a revision that addresses the issues, together with a cover letter explaining how you did so. My plan is to resend the revision to the present referees.

Reviewers' comments:

Reviewer's Responses to Questions

**Comments to the Author**

1. Is the manuscript technically sound, and do the data support the conclusions?

Reviewer #1: Yes

Reviewer #2: Partly

2. Has the statistical analysis been performed appropriately and rigorously? 

Reviewer #1: Yes

Reviewer #2: Yes

3. Have the authors made all data underlying the findings in their manuscript fully available?

Reviewer #1: Yes

Reviewer #2: Yes

4. Is the manuscript presented in an intelligible fashion and written in standard English?

Reviewer #1: Yes

Reviewer #2: Yes

5. Review Comments to the Author

Reviewer #1: In this paper, the authors study how the manipulation of visual experience through the use of virtual reality can affect the perception of pain due to movement in participants with non-specific neck pain. The main result of the study is to have identified a greater amplitude of pain-free movement when visual manipulation reduces the perceived rotation angle compared to when it is increased. The effect reached significance only within the group without kinesiophobia.

The article is written in appropriate and easily understandable language. Data analysis, the process is fully described and seems to have been conducted appropriately in relation to the characteristics of the research.

An interesting aspect is also the easy transportability in the clinical practice of these types of procedures. However, the literature on these aspects is more extensive than perhaps transpires from what is presented in the introduction and could benefit from slight enrichment.

At the discretion of the authors, I believe that some passages could benefit from brief clarification or elaboration.

The statements in the abstract should be made more explicit and understandable. In particular, results and discussion could clearly report not only that effects have been identified but also in what direction these are going (i.e. effects on participants’ painless movements).

56 The concept of "non-specific neck pain" should be better defined.

66-67 and Fig.1A. The authors refer to another study, but a figure in reference (6) depicts the same subjects with the same experimental setup with the same caption as fig 1A, this may be confusing. Please clarify in the text, photo/picture caption, or both if necessary.

The authors in their manuscript provide all the necessary information. However, for the convenience of the reader, it would seem to me to be useful to indicate in the participants section some basic info (which the authors actually include in the first few lines of the results) in order to understand with more immediacy numerosity and demographic characteristics of the participants.

L91 Please be a little clearer about the exclusion criterion. Is the problem in vision or possible problems related to wearing an HMD while wearing glasses? Was corrected to normal vision using contact lenses was ok?

Genders are differently represented, and in the rest of the manuscript, there is no mention of gender except in the table. Please include explicitly in the text, either in the participant section or in the results and/or discussion, the rationale for this choice and some consideration of the role of this variable in the study.

167-170 Appropriate pilot study was used to test the recognition effect of manipulation on specific stimuli. Although in non-identical procedures, visual perspective manipulation via virtual reality is nonetheless present in the literature, and providing one or two insights might be appropriate for the manuscript and useful to the reader.

186 It might be interesting for the reader to have one or two examples of environments (e.g., 1 natural and 1 artificial) available directly in the manuscript from among the supplementary materials.

Also, about the environments used, it seems to me that the choice of the types of such environments is not discussed except to avoid possible learning effects and memorization of visual references. Two environments are natural and very bright, while four are artificial and dark. The literature addresses these types of differences from numerous perspectives, however, given the short duration of the exposures and the type of task, it may not be of particular relevance. However, it would be appropriate to include in the manuscript, also in the discussion (around lines 344-347 would seem appropriate to me), some brief consideration and reference regarding this choice and the possible implications of the characteristic qualities of the environment.

Reviewer #2: General comment - The aim of this study is clinically relevant and it will be of interest to clinicians. However, to be even more clinically and scientifically relevant, the results must be reinterpreted according to current guidelines in terms of p values.

Line 2 – “Are people with non-specific neck pain”?

Abstract

Line 36 – Specify “people with non-specific neck pain” and add the comparison group (fear vs no fear).

Line 37 – Specify the population (e.g., acute, subacute, chronic)? It would be nice to have this specification for the whole manuscript.

Line 39 – Replace “impact” with “effect” to be consistent.

Line 44 – A p-value > 0.05 does not mean that there is no effect. Actually, the effect is η² = 0.044. This effect is really close to the other one (η² = 0.06) with a p-value < 0.05. I advise you to read the American Statistical Association statement about p-values. It is not recommended to focus on the “statistical significance” (p < 0.05) anymore.

Introduction

Line 56 – It may be interesting to add a definition of “non-specific neck pain”, because you included traumatic neck pain in the sample and this type of neck pain is not always included in the “non-specific neck pain” population.

Line 58 – “that may be perceived as threatening”?

Line 66 – Again, it is not right to state that there was no effect only based on p-values. In interpreting the results, we should include the exact p-value, the effect size, and its 95%CI. Furthermore, it should be better to show the effects found in that study and then compare them with the effect found in the other one. If there are differences in effect sizes, it can be very interesting to find some possible explanations.

Lines 74-75 – “Individuals who are not fearful of movement may unlearn the association made between movement and pain, allowing pain to extinguish”. This is a strong hypothesis, because it says that pain can be extinguished only because of the unlearned association. However, not all people with non-specific neck pain have fear of movement, but they still suffer from pain.

Lines 78-79 - Maybe replace with "the aim for the current study was to evaluate whether the effect of visual feedback manipulation is greater in people with non-specific neck pain that are fearful of movement, compared to people with non-specific neck pain that are not»? Or something like that. However, in both cases, your hypothesis seems to be a superiority one (“more susceptible”). It will have an impact on the statistical hypothesis and testing used.

Methods and materials

General comment – The manuscript does not follow best practices for reporting. In addition, there is no registered protocol of the experiment.

General comment - You separated people with fear into two subgroups (TSK and FABQ-pa). I understand, but as you said earlier, these are two closely related constructs of fear. So why did you not combine them together into one group (“fearful people”)? And after that, it can be divided into two subgroups to see if there are some differences in the effect size.

Line 84 – What was the sampling method used? Was it probabilistic (random) or non-probabilistic (e.g., convenience, snowballing)? It should be specified as it impacts the generalizability of the findings.

Line 95 – “Based on an expected effect size of ηp² = 0.145 (i.e., 0.29/2)”. What is the basis for that? Why did you divide the effect by 2? If there is no specific reason, you should specified that it was arbitrary.

Line 110 – “motion sickness was evaluated”. Why? What information did it provide? Is it related to the purpose of the study?

Line 132 – Please add a reference for the cut-off score of FABQ-pa.

Line 143 – I don’t understand why this outcome is assessed. I don’t think there is a link with the purpose of the study. Is motion sickness susceptible to influence the pain-free ROM or another outcome? If yes, it should be specified in the introduction.

Line 184 – Please specify the method of randomization.

Line 186 – There are six environments. So, one environment per repetition for each condition? It is not clear.

Line 194 – “different”. Here you say "different" (equivalence or not), but the aim of the study was to evaluate whether the effect was greater in fearful subjects, to see if they are more susceptible (superiority or not). The hypotheses are not the same, as well as the type of p-value (one-sided or two-sided).

Line 194 - It is stated that there are only 2 groups (with fear and without fear), however in the analyses there are 3 groups (without fear, fear with TSK, fear with FABQ-pa). It needs to be clarified.

Line 195 – Why is the absolute data analysis not included in the manuscript?

Line 202 - Thresholds are not clearly distinct, there are overlaps (e.g., 0.059 is small or medium?).

Line 205 – “two subgroups”. In total there are 3 subgroups. Maybe you want to say that there are 2 subgroups per analysis? If yes, please specify.

Results

General comment – Results need to be reinterpreted according to the American Statistical Association statement on p-values. We should not use the “statistical significance” (p<0.05) to conclude if there is an effect or not. Results should be described with exact p-values, effect size, and its 95%CI. Adding the relative changes of range of motion would be great, as it allows readers to better understand the changes observed and to judge if they may be of clinical relevance (effect sizes in terms of η² might be difficult to interpret for clinicians).

Line 218 – In the methods it is stated that 40 subjects were needed (according to the sample size calculation, 20 per group). However, 75 people participated. Why?

Lines 221 to 223 – If I understand, you included people with fear of physical activity into the "no fear" group (compared to TSK), and you included people with kinesiophobia into the "no fear" group (compared to FABQ). If yes, I am not sure this is right because they are fearful in both cases. I don't think they can be considered as "no fearful". Including people with kinesiophobia or fear of physical activity in the “no fear” group may induce biases in the results as they are fearful in both cases. So, you compared “fearful” people with “fearful + no fearful” people.

Line 223 – “10 participants had fear according to both questionnaires”. So, these participants were included twice in the analyses? Once for TSK and once for FABQ-pa?

Line 225 – “no differences”. Based on what? If you refer to the p-values, please indicate the exact p-value. However, it would be interesting to add the values of age and duration of neck pain for both groups. As mentioned earlier, the p-value alone is not sufficient to state that there is a difference or not.

Lines 226-227 – “more disabled, scored higher on pain intensity and was more limited in the pain-free range of rotation”. Based on what? If you refer to the p-values, please indicate the exact p-value for each comparison. However, it would be interesting to add the values of pain intensity, disability, and ROM for both groups. As mentioned earlier, the p-value alone is not sufficient to state that there is a difference or not.

Line 230 – What about people with fear of physical activity?

Line 243 – “Numeric” rather than “Nummeric”.

Line 255 – It is not recommended to use the “statistical significance” anymore. Please read the American Statistical Association statement on p-values. It would be great to show the relative change in range of motion and its 95%CI where appropriate. In addition of the effect size, it gives a better idea of the effect (it is well done in the S3 appendix). Interpreting these effects as clinically relevant or not would also be a good idea.

Lines 263 to 266 – Do not use statistical significance. For example, you could say that “differences in range of motion between the overstated gain and the control condition (p=0.051, ղ2=0.052), and between the understated gain and the control condition (p=0.077, ղ2=0.044) were a bit smaller than the effect found between the understated and the overstated conditions”. Or something like that, to focus not on statistical significance but on effect sizes. Adding a 95%CI for effect sizes would also be good.

Line 272 – “conflicting” rather than “conflciting”. Why is it “conflicting”? Is it because the p-value is superior to 0.05?

Line 280 – “Median” rather than “mean”?

Line 283 – You indicated 75 participants before, now it is 76. Please correct.

Line 287 – I still don’t understand why this outcome is evaluated and its usefulness in this study.

Discussion

Line 303 – Replace “pain-related” by “pain-free” to be consistent.

Lines 306-307 – Can these factors play a role in the effects observed? If yes, it should be discussed. You do not mention these factors as potential confounders in the differences in the effects observed.

Lines 321 to 325 – First, the absolute range of motion scores are detailed in the S3 Appendix, however I am not sure that readers will look at it. These scores are discussed here without any specific reason, as you mentioned earlier in the manuscript that you would use the relative data (to account for differences in the overall neck range of motion). Why do you discuss the absolute changes and not the relative ones? In the manuscript, you only show the analyses for the relative data, but in the discussion, you only mention the absolute data. It is not clear. Second, these differences in absolute range of motion are on the order of 4-5 degrees for the total range of motion in rotation. Is it clinically relevant? I think it would be good to specify the clinical relevance of these findings.

Line 338 – Why is it convincing? What is your basis for that?

Line 343 - Again, "no effect" means that the effect size equals 0. Was it the case? Or just a p-value > 0.05?

Lines 367-368 – The conclusion is a bit too ambitious. I advise you to interpret according to the results. Maybe something like "In people with non-specific neck pain, those with fear of movement may be more susceptible for the effect of visual feedback manipulation than people without fear of movement. Because the effect sizes are small to medium and because of the limitations mentioned, the results should be interpreted with caution”. Results should also be interpreted according to the sample characteristics (more females, middle-aged people, duration of neck pain, mean pain intensity, disability). Please revise it in the abstract too.

S3 Appendix

Line 4 - The Kolmogorov-Smirnov test is used to assess normality. However, in the manuscript it is stated that normality is assessed via Q-Q plots and histograms. Why is it different here?

Lines 13 to 17 – Interaction effect was present in both cases, however it was no “significant” based on the “p<0.05” threshold. Please see the American Statistical Association statement about p-values. These results need to be reinterpreted.

Lines 13 to 30 – Are these mean changes clinically relevant?

6. PLOS authors have the option to publish the peer review history of their article (what does this mean?). If published, this will include your full peer review and any attached files.

Reviewer #1: No

Reviewer #2: No

---

## [Author Response · Author response to Decision Letter 0]

1 May 2023

Accompanying response letter

Reviewer # 1

Comments to the Author

In this paper, the authors study how the manipulation of visual experience through the use of virtual reality can affect the perception of pain due to movement in participants with non-specific neck pain. The main result of the study is to have identified a greater amplitude of pain-free movement when visual manipulation reduces the perceived rotation angle compared to when it is increased. The effect reached significance only within the group without kinesiophobia.

The article is written in appropriate and easily understandable language. Data analysis, the process is fully described and seems to have been conducted appropriately in relation to the characteristics of the research.

An interesting aspect is also the easy transportability in the clinical practice of these types of procedures. However, the literature on these aspects is more extensive than perhaps transpires from what is presented in the introduction and could benefit from slight enrichment. 

At the discretion of the authors, I believe that some passages could benefit from brief clarification or elaboration.

Abstract

1. The statements in the abstract should be made more explicit and understandable. In particular, results and discussion could clearly report not only that effects have been identified but also in what direction these are going (i.e. effects on participants’ painless movements ).

Response: We agree with this comment. We adjusted this in the abstract and reported the direction of the effects (lines 47-50). 

2. Line 56 - The concept of "non-specific neck pain" should be better defined.

Response: Although this term is widely used, we can imagine that specifying the concept of 'non-specific neck pain' leads to more clarity. Therefore we defined it as ‘neckpain without a specific underlying pathology’1 (See line 39). A further specification of the characteristics of the participants is described in the selection criteria in the Methods section (See lines 93-100).

3. Lines 66-67 and Fig.1A. The authors refer to another study, but a figure in reference (6) depicts the same subjects with the same experimental setup with the same caption as fig 1A, this may be confusing. Please clarify in the text, photo/picture caption, or both if necessary.

Response: It is correct that we published a separate study using the same materials (e.g., VR headset, chair, fixation belt, computer) in an equivalent setting (as shown in the picture in figure 1A). However, this previously published article focused on different research questions, namely: (1) to determine whether pain-free range of motion increased and decreased when visual feedback understated or overstated true neck rotation with 20%, (2) to explore whether people with (long-lasting) chronic neck pain were more prone to the effect of visual feedback manipulation than people with subacute neck pain. This study did not address the effect of VR in people with fear of movement. So, the research questions from the current study generated from the first study, used different participants, different gain conditions (i.e., manipulation of the visual feedback with 30% rather than 20%) and different VR-environments. We clarified this in the picture (see figure 1). Despite similarities, the pictures differ from the pictures used in the previous publication. Taking into account point 8, we added two examples of the environments used to figure 1, and made this available directly in the manuscript. 

4. The authors in their manuscript provide all the necessary information. However, for the convenience of the reader, it would seem to me to be useful to indicate in the participants section some basic info (which the authors actually include in the first few lines of the results) in order to understand with more immediacy numerosity and demographic characteristics of the participants.

Response: We added the info as requested in the Methods (see line 91).

5. Line 91 - Please be a little clearer about the exclusion criterion. Is the problem in vision or possible problems related to wearing an HMD while wearing glasses? Was corrected to normal vision using contact lenses was ok.

Response: We have adopted the reviewer’s suggestion and added ‘(e.g., people who had poor vision when not wearing their glasses as they could not be worn in the VR condition. The use of contact lenses was not a problem)’ to the description of the exclusion criteria (see lines 99-100)

6. Genders are differently represented, and in the rest of the manuscript, there is no mention of gender except in the table. Please include explicitly in the text, either in the participant section or in the results and/or discussion, the rationale for this choice and some consideration of the role of this variable in the study.

Response: We assessed gender to describe the sample. Indeed, in this study genders are differently represented. Considering the fact that neck pain between the ages of 45 and 64 is almost 50% more common in women than in men2, we belief that the distribution within this study reflects the population of people with nonspecific neck pain. We added this interpretation to the text (see line 346-347). We did not look at differences between men and women regarding the influence of fear. Due to the limited sample size of the fear group in this study, it was not possible to determine an effect of gender.

7. Lines 167-170 Appropriate pilot study was used to test the recognition effect of manipulation on specific stimuli. Although in non-identical procedures, visual perspective manipulation via virtual reality is nonetheless present in the literature, and providing one or two insights might be appropriate for the manuscript and useful to the reader.

Response: We agree with the reviewer that insights regarding visual perspective manipulation via virtual reality is present in the literature. However, to our best knowledge, the sensitivity to rate of change in gains (applied by redirected walking) is studied in other conditions (e.g. upper limb movements, gait), but is not described related to neck movements (apart from the references that were already mentioned in the study). In our previous study the use of a 20% gain did not influence pain free range of motion3. We expected that a larger change in gain was needed to induce an effect. However, we did not know if we could enlarge the gain without being noticed. Therefore we performed the pilot study that is referred to.

8. Line 186- It might be interesting for the reader to have one or two examples of environments (e.g., 1 natural and 1 artificial) available directly in the manuscript from among the supplementary materials.

Response: Two examples of the environments used are added to Figure 1 (C and D)

9. Also, about the environments used, it seems to me that the choice of the types of such environments is not discussed except to avoid possible learning effects and memorization of visual references. Two environments are natural and very bright, while four are artificial and dark. The literature addresses these types of differences from numerous perspectives, however, given the short duration of the exposures and the type of task, it may not be of particular relevance. However, it would be appropriate to include in the manuscript, also in the discussion (around lines 344-347 would seem appropriate to me), some brief consideration and reference regarding this choice and the possible implications of the characteristic qualities of the environment.

Response: As indicated in the introduction, conflicting results have been found in previous studies on the effect of visual feedback on pain-free range of motion. As suggested by the reviewer, a possible explanation for the different results between the two previous studies can be that different VR-environments were used (i.e., a countryside, park, mountain, church grounds, dining room and living room by Harvie et al.4, and different locations in one forest-environment by Kragting et al.3). In this study, we chose to use similar (indoor and outdoor) environments as used in the study of Harvie, to make the studies comparable and avoid bias. We included this consideration in lines 199-200. 

Furthermore, a previous study that aimed to investigate the differential influence of multiple VR-environments on pain, reported that pain sensitivity was not modulated by context5. This information is used in the discussion (see line 377-378).

Reviewer: # 2

Comments to the Author

General comment - The aim of this study is clinically relevant and it will be of interest to clinicians. However, to be even more clinically and scientifically relevant, the results must be reinterpreted according to current guidelines in terms of p values.

Response: We agree with this reviewer’s remark. We added the confidence intervals in Table 2 and focused more on the effect sizes than on the statistical significance.

1. Line 2 – “Are people with non-specific neck pain”?

Response: We have incorporated this suggestion in the title.

Abstract

2. Line 36 – Specify “people with non-specific neck pain” and add the comparison group (fear vs no fear). 

Response: We have specified the neck pain group (see line 39). The comparison group is described in lines 42-45.

3. Line 37 – Specify the population (e.g., acute, subacute, chronic)? It would be nice to have this specification for the whole manuscript.

Response: We have specified the population regarding the stage of the disorder (acute, subacute, chronic, long-lasting chronic) in Table 1 and added the IPQ 25-75% regarding ‘duration of neck pain’. In our opinion, this information is too detailed for an abstract. We decided not to include this specification (acute, subacute, chronic, long-lasting chronic) in the whole manuscript, as our previous study showed that the duration of neck pain had no impact on the effect of visual feedback manipulation on pain-free range of motion3.

4. Line 39 – Replace “impact” with “effect” to be consistent.

Response: Thank you for this comment. We replaced it (line 42).

5. Line 44 – A p-value > 0.05 does not mean that there is no effect. Actually, the effect is η² = 0.044. This effect is really close to the other one (η² = 0.06) with a p-value < 0.05. I advise you to read the American Statistical Association statement about p-values. It is not recommended to focus on the “statistical significance” (p < 0.05) anymore.

Response: We agree with the reviewer’s remark. We added the confidence intervals in Table 2 and focused more on the effect sizes, rather than on the statistical significance.

Introduction

6. Line 56 – It may be interesting to add a definition of “non-specific neck pain”, because you included traumatic neck pain in the sample and this type of neck pain is not always included in the “non-specific neck pain” population.

Response: We defined non-specific neck pain as ‘neck pain without a specific underlying pathology’1 (See line 61). A further specification of the characteristics of the participants is described in the selection criteria in the Methods section. (See lines 93-100).

7. Line 58 – “that may be perceived as threatening”?

Response: Thank you for this comment. We adjusted the sentence (line 63).

8. Line 66 – Again, it is not right to state that there was no effect only based on p-values. In interpreting the results, we should include the exact p-value, the effect size, and its 95%CI. Furthermore, it should be better to show the effects found in that study and then compare them with the effect found in the other one. If there are differences in effect sizes, it can be very interesting to find some possible explanations.

Response: In line with the general comment, the interpretation of the results of the studies we referred to in the introduction are based on the exact p-values, effect sizes and its 95%CI. In our opinion, it is unusual to include the explicit results of other studies in the introduction. A possible explanation for the different results between the two studies is given in the next paragraph (see lines 75-76). 

9. Lines 74-75 – “Individuals who are not fearful of movement may unlearn the association made between movement and pain, allowing pain to extinguish”. This is a strong hypothesis, because it says that pain can be extinguished only because of the unlearned association. However, not all people with non-specific neck pain have fear of movement, but they still suffer from pain.

Response: We agree with the reviewer that in people with non-specific neck pain, there can be multiple reasons for the persistence of pain. In this study, we started from the hypothesis that pain may, partially, be a learned experience. We then propose a possible explanation for this hypothesis. To emphasize that it is an assumption, we have made the following adjustment ‘The underlying assumption is that individuals who are not fearful of movement may unlearn the association made between movement and pain, allowing pain to extinguish’ (see lines 80-81).

10. Lines 78-79 - Maybe replace with "the aim for the current study was to evaluate whether the effect of visual feedback manipulation is greater in people with non-specific neck pain that are fearful of movement, compared to people with non-specific neck pain that are not»? Or something like that. However, in both cases, your hypothesis seems to be a superiority one (“more susceptible”). It will have an impact on the statistical hypothesis and testing used.

Response: We agree with the reviewer that the objective of the study should have been formulated neutrally. We've adjusted this (see lines 84-86).

Methods and materials

General comment – The manuscript does not follow best practices for reporting. In addition, there is no registered protocol of the experiment.

Response: We followed the guidelines for reporting, however, we confirm that the protocol for the study was not published or registered.

General comment - You separated people with fear into two subgroups (TSK and FABQ-pa). I understand, but as you said earlier, these are two closely related constructs of fear. So why did you not combine them together into one group (“fearful people”)? And after that, it can be divided into two subgroups to see if there are some differences in the effect size.

Response: This comment of the reviewer made us reflect on our analyses. We formulated three different options how to modify our paper and sought advice from the Editor and reviewer (see Author Query 30-3-2023). Based on the advice received, we re-analysed our data (see comment 24).

11. Line 84 – What was the sampling method used? Was it probabilistic (random) or non-probabilistic (e.g., convenience, snowballing)? It should be specified as it impacts the generalizability of the findings .

Response: We added the sampling method used (see lines 92).

12. Line 95 – “Based on an expected effect size of ηp² = 0.145 (i.e., 0.29/2)”. What is the basis for that? Why did you divide the effect by 2? If there is no specific reason, you should specified that it was arbitrary. 

Response: The design of the reference study (within comparisons)4 differed from the design of the current study (between-within), therefore we anticipated a smaller effect size. Indeed, the choice to divide the expected effect size by two was arbitrary. We specified this in the text (see lines 102-105).

13. Line 110 – “motion sickness was evaluated”. Why? What information did it provide? Is it related to the purpose of the study?

Response: We monitored motion sickness in this study. Motion sickness is a common side effect in VR, especially in people with neck pain and a possible barrier when considering implementing VR in clinical practice (see lines 119-121). Furthermore, in this study we increased the gain compared to other studies (i.e., manipulation of the visual feedback with 30% rather than 20%). In our opinion it is important to monitor this possible side effect, because it is a frequently mentioned phenomenon.

14. Line 132 – Please add a reference for the cut-off score of FABQ-pa.

Response: We added a reference for the cut-off score of the FABQpa (see line 141).

15. Line 143 – I don’t understand why this outcome is assessed. I don’t think there is a link with the purpose of the study. Is motion sickness susceptible to influence the pain-free ROM or another outcome? If yes, it should be specified in the introduction.

Response: Indeed, we have chosen not to include the evaluation of motion sickness in the objectives of the study. In our opinion this was a minor aim. An explanation for the choice made is given in comment 13.

16. Line 184 – Please specify the method of randomization.

Response: We specified the method of randomization in line 193-195.

17. Line 186 – There are six environments. So, one environment per repetition for each condition? It is not clear.

Response: To clarify the procedure we added ‘Hence, one environment was used per repetition for each condition’ to the sentence in line 198-199. 

18. Line 194 – “different”. Here you say "different" (equivalence or not), but the aim of the study was to evaluate whether the effect was greater in fearful subjects, to see if they are more susceptible (superiority or not). The hypotheses are not the same, as well as the type of p-value (one-sided or two-sided ).

Response: As suggested by the reviewer in comment 10, we have reformulated our aim (see lines 84-86). This is in line with the formulation used in this section.

19. Line 194 - It is stated that there are only 2 groups (with fear and without fear), however in the analyses there are 3 groups (without fear, fear with TSK, fear with FABQ-pa). It needs to be clarified.

Response: In accordance with the advice, we performed 2 analyses with 2 subgroups per analysis (i.e., ‘no fear on both scales’ (TSK≤37 & FABQ-pa≤14) compared with ‘fearful for movement- kinesiophobia’ (TSK>37); and ‘no fear on both scales’ compared to ‘fear for physical activity’(FABQ-pa>14)). This is clarified in the text (see lines 206-211).

20. Line 195 – Why is the absolute data analysis not included in the manuscript ?

Response: As reported in the manuscript, we chose to report the relative data analysis in the main manuscript, because we wanted to anticipate expected differences between subjects (especially between people with and without fear of movement). Indeed, in the current sample the mean range of rotation differed between the people with and without fear of movement. To account for differences in the overall neck range of motion between subjects, for each participant, the data from the overstated and understated condition were transformed to a proportion of the mean range of rotation in the control condition (i.e., relative data). We have chosen not to include the absolute data-analysis in the main manuscript to avoid an overload of results, and thus confusion. Based on the reviewer's comment, we have decided to include the results of the absolute analyses in Table 2. We think this will support the clinician in interpreting the results.

21. Line 202 - Thresholds are not clearly distinct, there are overlaps (e.g., 0.059 is small or medium ?).

Response: Thank you for this comment. We made an adjustment (see lines 214-215).

22. Line 205 – “two subgroups”. In total there are 3 subgroups. Maybe you want to say that there are 2 subgroups per analysis? If yes, please specify.

Response: Thank you for this comment. We specified this (see line 217).

Results

General comment – Results need to be reinterpreted according to the American Statistical Association statement on p-values. We should not use the “statistical significance” (p<0.05) to conclude if there is an effect or not. Results should be described with exact p-values, effect size, and its 95%CI. Adding the relative changes of range of motion would be great, as it allows readers to better understand the changes observed and to judge if they may be of clinical relevance (effect sizes in terms of η² might be difficult to interpret for clinicians).

Results are described with exact p-values and effect sizes. Both the absolute and relative changes in range of motion (including their 95%CI) are added in Table 2. We agree with the reviewer that this will support readers to better understand the changes observed and to judge if they may be of clinical relevance.

23. Line 218 – In the methods it is stated that 40 subjects were needed (according to the sample size calculation, 20 per group). However, 75 people participated. Why ?

Response: The subgroups (fear/no fear) were not equally distributed. We continued to enrol participants until we could include about 20 people with fear of movement.

24. Lines 221 to 223 – If I understand, you included people with fear of physical activity into the "no fear" group (compared to TSK), and you included people with kinesiophobia into the "no fear" group (compared to FABQ). If yes, I am not sure this is right because they are fearful in both cases. I don't think they can be considered as "no fearful". Including people with kinesiophobia or fear of physical activity in the “no fear” group may induce biases in the results as they are fearful in both cases. So, you compared “fearful” people with “fearful + no fearful” people .

Response: We agree with the reviewer that a positive score on one of the two scales may indicate a form of fear, which may have biased the results. Classifying the people who scored positive on the TSK, but negative on the FABQ-pa, in the 'no-fear of physical activity group’ probably made the contrast between the two groups smaller. The same applies to the analysis based on the TSK. In accordance with the advice from the reviewer (see email dd. 06-04-2023), we reanalysed the data and only included people who scored negative on both scales (TSK≤37 &FABQ-pa≤ 14) in the no fear group. We then also added an exploratory analysis in which people who were negative on both scales (N=46) were compared to participants who scored positive on both scales (N=10). The effect size is then twice as large (p=0.007, ղ2=0.112), confirming the impact of fear.

25. Line 223 – “10 participants had fear according to both questionnaires”. So, these participants were included twice in the analyses? Once for TSK and once for FABQ-pa ?

Response: as mentioned in our response in comment 19 & 24 we performed 2 separate analyses with 2 subgroups per analysis (i.e., ‘no fear on both scales’ (TSK≤37 & FABQ-pa≤14) compared with ‘fearful for movement- kinesiophobia’ (TSK>37); and ‘no fear on both scales’ compared to ‘fear for physical activity’(FABQ-pa>14)). So indeed, most participants were included twice in the analyses. As mentioned in comment 24 we also added an exploratory analysis (to enlarge the contrasts between the subgroups) in which people who were negative on both scales (N=46) were compared to participants who scored positive on both scales (N=10).

26. Line 225 – “no differences”. Based on what? If you refer to the p-values, please indicate the exact p-value. However, it would be interesting to add the values of age and duration of neck pain for both groups. As mentioned earlier, the p-value alone is not sufficient to state that there is a difference or not .

Response: To compare the participants characteristics between the subgroups, we performed statistical tests (independent sample t-tests or (in case of non-normality of the data) non-parametric Mann-Whitney tests). We chose to report values of age, duration, pain intensity, disability, kinesiophobia, fear of physical activity and ROM, including the results of the statistical tests, in Table 1 and not in the text, as these results do not answer the main research question and we intended to avoid duplication. Therefore, we suggest referring to the table for these data (as done in the original version)

27. Lines 226-227 – “more disabled, scored higher on pain intensity and was more limited in the pain-free range of rotation”. Based on what? If you refer to the p-values, please indicate the exact p-value for each comparison. However, it would be interesting to add the values of pain intensity, disability, and ROM for both groups. As mentioned earlier, the p-value alone is not sufficient to state that there is a difference or not.

Response: We specified in the text that ‘Table 1 provides an overview of the participant characteristics per group including the results from the comparisons between subgroups’ (see line 239-240).

28. Line 230 – What about people with fear of physical activity ?

Response: We added the result regarding mean ROM in people with fear of physical activity (see line 243).

29. Line 243 – “Numeric” rather than “Nummeric”.

Response: Thank you for this comment. We made an adjustment (see line 254).

30. Line 255 – It is not recommended to use the “statistical significance” anymore. Please read the American Statistical Association statement on p-values. It would be great to show the relative change in range of motion and its 95%CI where appropriate. In addition of the effect size, it gives a better idea of the effect (it is well done in the S3 appendix). Interpreting these effects as clinically relevant or not would also be a good idea.

Response: The exact p-values and effect sizes are described and the effect sizes are interpreted. The 95% CI are expressed in Figure 2 and Table 2. The absolute values (mean ROM and 95%CI) have been added in Table 2. This has informative value for the clinician. 

31. Lines 263 to 266 – Do not use statistical significance. For example, you could say that “differences in range of motion between the overstated gain and the control condition (p=0.051, ղ2=0.052), and between the understated gain and the control condition (p=0.077, ղ2=0.044) were a bit smaller than the effect found between the understated and the overstated conditions”. Or something like that, to focus not on statistical significance but on effect sizes. Adding a 95%CI for effect sizes would also be good.

Response: As previously mentioned: results are described with exact p-values and effect sizes. Effect sizes are interpreted, and both the absolute as relative changes in range of motion (including their 95%CI) are added in Table 2. 

32. Line 272 – “conflicting” rather than “conflciting”. Why is it “conflicting”? Is it because the p-value is superior to 0.05? 

Response: We agree with the reviewer that the result of our prior analyses were not conflicting. We only found a small effect in the TSK group. The results of the recent analyses are in line with each other.

33. Line 280 – “Median” rather than “mean”?

Response: Thank you for this comment. Following Field (3th ed, p. 580)6, we have to report the test statistic, degrees of freedom and its significance. We made an adjustment (see line 315).

34. Line 283 – You indicated 75 participants before, now it is 76. Please correct.

Response: Thank you for this comment. We made an adjustment (see line 318).

35. Line 287 – I still don’t understand why this outcome is evaluated and its usefulness in this study.

Response: See response in comment 13

Discussion

36. Line 303 – Replace “pain-related” by “pain-free” to be consistent.

Response: Thank you for this comment. We checked the manuscript for the use of “pain-free”

37. Lines 306-307 – Can these factors play a role in the effects observed? If yes, it should be discussed. You do not mention these factors as potential confounders in the differences in the effects observed.

Response: It was expected that the differences in pain-free range of motion between the groups with and without fear of movement would influence the scores in the absolute data-analyses. Therefore the relative range of motion scores were used in the main analyses. This is already explained in the text (lines 187-191). It was not expected that the scores for pain and dysfunction were potential confounders, as a within design is used to test the effect of feedback manipulation on pain-free range of motion and the scores for pain and functioning were the same for each participant in the three gain conditions.

38. Lines 321 to 325 – First, the absolute range of motion scores are detailed in the S3 Appendix, however I am not sure that readers will look at it. These scores are discussed here without any specific reason, as you mentioned earlier in the manuscript that you would use the relative data (to account for differences in the overall neck range of motion). Why do you discuss the absolute changes and not the relative ones? In the manuscript, you only show the analyses for the relative data, but in the discussion, you only mention the absolute data . It is not clear. Second, these differences in absolute range of motion are on the order of 4-5 degrees for the total range of motion in rotation. Is it clinically relevant? I think it would be good to specify the clinical relevance of these findings.

Response: The absolute range of motion scores are easier to interpret, while the relative scores have no clear meaning for clinicians. However, we agree with the reviewer that it is strange to discuss the absolute data while in the rest of the manuscript the relative data were described. Therefore, this part is moved to the appendix. To make the relative data easier to interpret for clinicians, the change in pain-free range of motion is expressed in a percentage increase or decrease in range of motion.

39. Line 338 – Why is it convincing? What is your basis for that?

Response: This part of the text has been omitted in the revised version of the discussion

40. Line 343 - Again, "no effect" means that the effect size equals 0. Was it the case? Or just a p-value > 0.05? 

Response: In this case we think we can say there is ‘no effect’, as the effect size is small, and the p-value high (p=0.133, ղp2=0.031).

41. Lines 367-368 – The conclusion is a bit too ambitious. I advise you to interpret according to the results. Maybe something like "In people with non-specific neck pain, those with fear of movement may be more susceptible for the effect of visual feedback manipulation than people without fear of movement. Because the effect sizes are small to medium and because of the limitations mentioned, the results should be interpreted with caution”. Results should also be interpreted according to the sample characteristics (more females, middle-aged people, duration of neck pain, mean pain intensity, disability). Please revise it in the abstract too.

Response: We believe that the first statement (‘This experiment supports the view that pain-free ROM can be influenced by visual perception of the amount of rotation (i.e., the main effect of gain, independent of the presence of fear)’) can be well substantiated by the results (TSK: p<0.001, ղp2=0.158; FABQpa: p<0.001, ղp2=0.195) -> large effect sizes. Given the medium effect sizes in both the analyses regarding the influence of fear (i.e., the interaction effect between fear of movement and visual feedback manipulation), we think that the interpretation that ‘People with fear of movement seem to be more susceptible for the effect of visual feedback manipulation than people without fear of movement’, is sufficiently consistent with the results (see lines 410-413).

S3 Appendix

42. Line 4 - The Kolmogorov-Smirnov test is used to assess normality. However, in the manuscript it is stated that normality is assessed via Q-Q plots and histograms. Why is it different here? 

Response: We used the Q-Q plots, the histograms and the KS test to consider if the data were sufficiently normally distributed, so we have adjusted this in the text.

43. Lines 13 to 17 – Interaction effect was present in both cases, however it was no “significant” based on the “p<0.05” threshold. Please see the American Statistical Association statement about p-values. These results need to be reinterpreted.

Response: In accordance with the advice (see comment 24 and 25) we also re-analysed the data using the absolute range of motion scores. We performed 2 analyses with 2 subgroups per analysis (i.e., ‘no fear on both scales’ (TSK≤37 & FABQ-pa≤14) compared with ‘fearful for movement- kinesiophobia’ (TSK>37); and ‘no fear on both scales’ compared to ‘fear for physical activity’(FABQ-pa>14)). The analyses and results are reinterpreted, following the American Statistical Association statement. The confidence intervals are expressed in Table 2 and S3 Figure.

44. Lines 13 to 30 – Are these mean changes clinically relevant ?

Response: The mean changes in the ‘no fear’ group do not seem to be clinically relevant (as also discussed in the main manuscript (lines 390-398), while the changes in pain-free range of motion in the ‘fear of physical activity group’ and the ‘kinesiophobia’ group, might be clinically relevant. This is explained in lines 401-407.

References

1. Tsakitzidis G, Remmen R, Peremans L, et al. Non-specific neck pain: diagnosis and treatment. Good Clinical Practice (GCP) KCE Reports C. 2009;119.

2. Bier JD, Scholten-Peeters, G.G.M., Staal, J.B., Pool, J., Tulder, M. van, Beekman, E., Meerhoff, G.M. Knoop, J., Verhagen, A.P., . KNGF richtlijn nekpijn. In. Praktijkrichtlijn. Amersfoort: KNGF; 2016.

3. Kragting M, Schuiling SF, Voogt L, Pool-Goudzwaard AL, Coppieters MW. Using Visual Feedback Manipulation in Virtual Reality to Influence Pain-Free Range of Motion in People with Nonspecific Neck Pain. Pain Pract. 2020.

4. Harvie DS, Broecker M, Smith RT, Meulders A, Madden VJ, Moseley GL. Bogus visual feedback alters onset of movement-evoked pain in people with neck pain. Psychol Sci. 2015;26(4):385-392.

5. Smith A, Carlow K, Biddulph T, Murray B, Paton M, Harvie DS. Contextual modulation of pain sensitivity utilising virtual environments. Br J Pain. 2017;11(2):71-80.

6. Field A. Discovering Statistics Using SPSS. Third ed. London: Sage Publications Ltd; 2009.

---

## [Decision Letter · Decision Letter 1]

29 May 2023

PONE-D-22-28987R1Visual feedback manipulation in virtual reality to influence pain-free range of motion. Are people with non-specific neck pain who are fearful of movement more susceptible?PLOS ONE

Dear Dr. Kragting,

Thank you for submitting your manuscript to PLOS ONE. After careful consideration, we feel that it has merit but does not fully meet PLOS ONE’s publication criteria as it currently stands. Therefore, we invite you to submit a revised version of the manuscript that addresses the points raised during the review process.

We look forward to receiving your revised manuscript.

Kind regards,

Mariella Pazzaglia

Academic Editor

PLOS ONE

Journal Requirements:

Additional Editor Comments:

As you can see, the reviewer thinks your paper is potentially publishable but that some changes would be required.

Reviewers' comments:

Reviewer's Responses to Questions

**Comments to the Author**

1. If the authors have adequately addressed your comments raised in a previous round of review and you feel that this manuscript is now acceptable for publication, you may indicate that here to bypass the “Comments to the Author” section, enter your conflict of interest statement in the “Confidential to Editor” section, and submit your "Accept" recommendation.

Reviewer #2: (No Response)

2. Is the manuscript technically sound, and do the data support the conclusions?

Reviewer #2: Yes

3. Has the statistical analysis been performed appropriately and rigorously? 

Reviewer #2: Yes

4. Have the authors made all data underlying the findings in their manuscript fully available?

Reviewer #2: Yes

5. Is the manuscript presented in an intelligible fashion and written in standard English?

Reviewer #2: Yes

6. Review Comments to the Author

Reviewer #2: Introduction

Lines 80-81 – The authors said “to emphasize that it is an assumption, we have made the following adjustment ‘The underlying assumption is that individuals who are not fearful of movement may unlearn the association made between movement and pain, allowing pain to extinguish’”, in relation to the comment number 9 (first revision). It's clearer that this is a hypothesis. However, my comment was more about the "allowing pain to extinguish" part. By saying this, the authors are assuming that the association between movement and pain is the only factor that determines the presence of pain. I'm not so sure about that. I think pain can be reduced, but not necessarily extinguished. I would suggest being more nuanced in the sentence and saying, for example, "allowing pain to be reduced".

Methods and materials

General comment – The authors have confirmed that they have followed reporting guidelines, but these are not mentioned in the manuscript. Please specify which reporting guidelines were followed in this manuscript.

Line 91 – I think that the first part of the sentence should be placed at the beginning of the results. The number of subjects is also written at line 238, but with different characteristics in brackets. It should be better to combine these two sentences in one (only the number of subjects and their characteristics) to avoid repetition.

Lines 119-121 – “Since motion sickness is a common side effect in VR (17), especially in people with neck pain (18), and a possible barrier when considering implementing VR in clinical practice, it was evaluated after the VR-experiment using the short version of the Misery Scale (sMISC)”.

Line 131 – “(FABQpa)” rather than “(FABQpa))”.

Line 132 – “Cut-off” rather than “cut-of”.

Line 212 – To be as complete and transparent as possible, I would suggest specifying that you will present results using exact p-values, the effect estimate and its 95%CI. Avoiding the use of the threshold "p<0.05" is not yet widespread in musculoskeletal research, so readers may be confused and wonder why you don't specify whether the results are statistically significant or not, if this is not stated here.

Results

Line 238 – My comment was: “In the methods it is stated that 40 subjects were needed (according to the sample size calculation, 20 per group). However, 75 people participated. Why?”. Authors responded: “The subgroups (fear/no fear) were not equally distributed. We continued to enroll participants until we could include about 20 people with fear of movement.”. I understand, but why didn't you keep the first 20 subjects with no fear of movement and only continue to enroll subjects with fear of movement? In any case, it's already been done. So, I suggest clarifying in the manuscript what you mentioned above, so that the reader is not confused about the difference between the sample size calculation and the final sample size.

Lines 244-245 – It says "In both groups" at the beginning of the sentence, but don't you mean "In both analyses"? I think it is not clear enough, because you have two fearful subgroups.

Line 250 – Add “pain-free” before “range of rotation” to be consistent.

Line 251 – Add “degrees” after “… with fear of physical activity 106.9 (38.4)”, to be consistent.

Table 1 – I don't think it's necessary to specify both sexes. In general, we only specify one sex (female or male), because if we know the numbers for one, we indirectly know the numbers for the other.

Line 265 – If you use a specific threshold for significance for these tests, you should specify which one you are using.

Lines 281-285 – At the beginning of the sentence (starting with “Contrasts…”) you compare pain free range of motion between the understated condition and the control condition, in fearful people, but in the next part of the sentence it is less clear. Do you talk about overstated versus control condition in fearful and non-fearful people, or do you compare fearful vs non fearful people in both conditions?

Line 282 – If you avoid using a specific threshold for statistical significance, it is best to avoid terms like “significantly”.

Lines 287-289 – The following part of the sentence is not clear for me: “decreased by 4.5%, 95%CI [-9.0%, 0.0%] (TSK) respectively 6.5%, 95%CI [-9.9%, -3.1%] (FABQpa) in the overstated condition.”. Could you rewrite it more clearly?

Line 289 – “This direction of the effect of visual feedback was as expected”. For me, this should be placed in the discussion.

Lines 289-290 – I had to reread the previous sentences to understand what the "overall effect" was. I think it would be a good idea to add in brackets what exactly it is.

Lines 324-325 – Same comment as for lines 289-290.

Lines 340-342 – For completeness and consistency, I would add the number of subjects with each type of symptom, in addition to the percentage (which is done in the previous section).

Discussion

Line 353 – What do you mean by “susceptibility to visual feedback manipulation”? Maybe “susceptibility for the effect of visual feedback manipulation”?

Lines 388-389 – My comment was: “Again, "no effect" means that the effect size equals 0. Was it the case? Or just a p-value > 0.05?”. Authors responded: “In this case we think we can say there is ‘no effect’, as the effect size is small, and the p-value high (p=0.133, ղp2=0.031)”. I don’t agree. To be consistent with the rest of the manuscript, authors should replace “no effect” with “small effect”.

Line 390 – Add “of” after “fear”.

Lines 408-421 – If you want to compare the fearful and non-fearful groups in the same way, you need to use the same relative changes. For non-fearful subjects, you're talking only about the change between the overstated condition and the control condition, whereas for fearful subjects, you're talking about the overall relative change. So, there's a big difference between the two groups.

Lines 426-427 – I would add “neck” before “pain-free range of motion” and specify that it is about rotation only. Sometimes you write “pain-free range of motion”, sometimes “cervical pain-free range of motion” and sometimes “neck pain-free range of motion”. Try to be consistent so that things are always written in the same way. In addition, I would clarify the population again. For example, the new sentence would be: “This experiment supports the view that neck pain-free range of motion in rotation can be influenced by visual perception of the amount of rotation, in people with non-specific neck pain”.

7. PLOS authors have the option to publish the peer review history of their article (what does this mean?). If published, this will include your full peer review and any attached files.

Reviewer #2: No

---

## [Author Response · Author response to Decision Letter 1]

8 Jun 2023

Author response letter

Journal Requirements:

Comments to the Author

Response: We reviewed our reference list to ensure that it is complete and correct. 

Reviewer: # 2

Comments to the Author

Introduction

1. Lines 80-81 – The authors said “to emphasize that it is an assumption, we have made the following adjustment ‘The underlying assumption is that individuals who are not fearful of movement may unlearn the association made between movement and pain, allowing pain to extinguish’”, in relation to the comment number 9 (first revision). It's clearer that this is a hypothesis. However, my comment was more about the "allowing pain to extinguish" part. By saying this, the authors are assuming that the association between movement and pain is the only factor that determines the presence of pain. I'm not so sure about that. I think pain can be reduced, but not necessarily extinguished. I would suggest being more nuanced in the sentence and saying, for example, "allowing pain to be reduced".

Response: We followed the suggestion and changed ‘extinguished’ to ‘reduced’ (lines 82-83)

Methods and materials

2. General comment – The authors have confirmed that they have followed reporting guidelines, but these are not mentioned in the manuscript. Please specify which reporting guidelines were followed in this manuscript.

Response: We added the reporting guideline to the manuscript (lines 111-112). Furthermore, we specified the study design in the Abstract (line 39) and we added the recruitment period to the Methods (line 92).

3. Line 91 – I think that the first part of the sentence should be placed at the beginning of the results. The number of subjects is also written at line 238, but with different characteristics in brackets. It should be better to combine these two sentences in one (only the number of subjects and their characteristics) to avoid repetition.

Response: We have followed this suggestion and removed the duplication (lines 242-243)

4. Lines 119-121 – “Since motion sickness is a common side effect in VR (17), especially in people with neck pain (18), and a possible barrier when considering implementing VR in clinical practice, it was evaluated after the VR-experiment using the short version of the Misery Scale (sMISC)”.

Response: Thank you for your suggestion. Given the length of the sentence, we chose to split the sentence (lines 121-123)

5. Line 131 – “(FABQpa)” rather than “(FABQpa))”.

Response: Corrected as suggested (line 133)

6 Line 132 – “Cut-off” rather than “cut-of”.

Response: Corrected as suggested (line 134)

7. Line 212 – To be as complete and transparent as possible, I would suggest specifying that you will present results using exact p-values, the effect estimate and its 95%CI. Avoiding the use of the threshold "p<0.05" is not yet widespread in musculoskeletal research, so readers may be confused and wonder why you don't specify whether the results are statistically significant or not, if this is not stated here.

Response: We agree with the reviewer’s suggestion. We specified this in the manuscript (lines 224-225).

Results

8 Line 238 – My comment was: “In the methods it is stated that 40 subjects were needed (according to the sample size calculation, 20 per group). However, 75 people participated. Why?”. Authors responded: “The subgroups (fear/no fear) were not equally distributed. We continued to enroll participants until we could include about 20 people with fear of movement.”. I understand, but why didn't you keep the first 20 subjects with no fear of movement and only continue to enroll subjects with fear of movement? In any case, it's already been done. So, I suggest clarifying in the manuscript what you mentioned above, so that the reader is not confused about the difference between the sample size calculation and the final sample size.

Response: We followed the reviewer’s comment and incorporated this suggestion (lines 244-246)

9. Lines 244-245 – It says "In both groups" at the beginning of the sentence, but don't you mean "In both analyses"? I think it is not clear enough, because you have two fearful subgroups.

Response: Thank you for identifying this. Changed as suggested. (lines 250)

10. Line 250 – Add “pain-free” before “range of rotation” to be consistent.

Response: Changed as suggested (line 256) and throughout the manuscript.

11. Line 251 – Add “degrees” after “… with fear of physical activity 106.9 (38.4)”, to be consistent.

Response: Changed as suggested (line 257)

12. Table 1 – I don't think it's necessary to specify both sexes. In general, we only specify one sex (female or male), because if we know the numbers for one, we indirectly know the numbers for the other.

Response: Changed as suggested and incorporated this in Table 1.

13. Line 265 – If you use a specific threshold for significance for these tests, you should specify which one you are using.

Response: Changed as suggested - We added whether we used t-test or Mann-Whitney U tests in Table 1.

14. Lines 281-285 – At the beginning of the sentence (starting with “Contrasts…”) you compare pain free range of motion between the understated condition and the control condition, in fearful people, but in the next part of the sentence it is less clear. Do you talk about overstated versus control condition in fearful and non-fearful people, or do you compare fearful vs non fearful people in both conditions?

Response: The contrasts reveal the differences between people with and without fear of movement in the understated condition compared to the control condition, and the differences between people with and without fear of movement in the overstated condition compared to the control condition. We modified the text to make this clear (lines 288-293).

15. Line 282 – If you avoid using a specific threshold for statistical significance, it is best to avoid terms like “significantly”.

Response: We agree with the reviewer. Changed as suggested. (line 289) 

16. Lines 287-289 – The following part of the sentence is not clear for me: “decreased by 4.5%, 95%CI [-9.0%, 0.0%] (TSK) respectively 6.5%, 95%CI [-9.9%, -3.1%] (FABQpa) in the overstated condition.”. Could you rewrite it more clearly?

Response: We modified the text and wrote this part more clearly (see lines 293-300).

17. Line 289 – “This direction of the effect of visual feedback was as expected”. For me, this should be placed in the discussion.

Response: We agree and adjusted the manuscript accordingly (lines 391-393).

18. Lines 289-290 – I had to reread the previous sentences to understand what the "overall effect" was. I think it would be a good idea to add in brackets what exactly it is.

Response: Changed as suggested: we added “(i.e., the difference in pain-free range of motion between the understated and the overstated condition)” (lines 300-303)

19. Lines 324-325 – Same comment as for lines 289-290.

Response: Changed as above (lines 337-339)

20. Lines 340-342 – For completeness and consistency, I would add the number of subjects with each type of symptom, in addition to the percentage (which is done in the previous section).

Response: Unlike for the ‘previous section’, these percentages are already mentioned in Table 3. We prefer not to follow the suggestion of the Reviewer to avoid duplication.

Discussion

21. Line 353 – What do you mean by “susceptibility to visual feedback manipulation”? Maybe “susceptibility for the effect of visual feedback manipulation”?

Response: We have incorporated this suggestion (line 366).

22. Lines 388-389 – My comment was: “Again, "no effect" means that the effect size equals 0. Was it the case? Or just a p-value > 0.05?”. Authors responded: “In this case we think we can say there is ‘no effect’, as the effect size is small, and the p-value high (p=0.133, ղp2=0.031)”. I don’t agree. To be consistent with the rest of the manuscript, authors should replace “no effect” with “small effect”.

Response: Changed as suggested (line 403).

23. Line 390 – Add “of” after “fear”.

Response: Changed as suggested (line 406).

24. Lines 408-421 – If you want to compare the fearful and non-fearful groups in the same way, you need to use the same relative changes. For non-fearful subjects, you're talking only about the change between the overstated condition and the control condition, whereas for fearful subjects, you're talking about the overall relative change. So, there's a big difference between the two groups.

Response: We agree with the reviewer that when comparing the fearful and the non-fearful groups in the same way, we have to compare the relative changes in the same conditions. We therefore compared the overall relative change in both groups in the revised version of the manuscript (lines 424-430).

25. Lines 426-427 – I would add “neck” before “pain-free range of motion” and specify that it is about rotation only. Sometimes you write “pain-free range of motion”, sometimes “cervical pain-free range of motion” and sometimes “neck pain-free range of motion”. Try to be consistent so that things are always written in the same way. In addition, I would clarify the population again. For example, the new sentence would be: “This experiment supports the view that neck pain-free range of motion in rotation can be influenced by visual perception of the amount of rotation, in people with non-specific neck pain”.

Response: Changed as suggested. We preferred to use ‘cervical’ instead of ‘neck’. (lines 444-445)

---

## [Editor Report · Decision Letter 2]

15 Jun 2023

Visual feedback manipulation in virtual reality to influence pain-free range of motion. Are people with non-specific neck pain who are fearful of movement more susceptible?

PONE-D-22-28987R2

Dear Dr. Maaike Kragting,

We’re pleased to inform you that your manuscript has been judged scientifically suitable for publication and will be formally accepted for publication once it meets all outstanding technical requirements.

Kind regards,

Mariella Pazzaglia

Academic Editor

PLOS ONE
---

## [Editor Report · Acceptance letter]

22 Jun 2023

PONE-D-22-28987R2 

Visual feedback manipulation in virtual reality to influence pain-free range of motion. Are people with non-specific neck pain who are fearful of movement more susceptible? 

Dear Dr. Coppieters:

I'm pleased to inform you that your manuscript has been deemed suitable for publication in PLOS ONE. Congratulations! Your manuscript is now with our production department. 

Kind regards, 

on behalf of

Dr. Mariella Pazzaglia 

Academic Editor

PLOS ONE